# Feasibility Study of Scanning Spectral Imaging Based on a Birefringence Flat Plate

**DOI:** 10.3390/s24103092

**Published:** 2024-05-13

**Authors:** Ilan Gadasi, Yoel Arieli

**Affiliations:** The Applied Physics/Electro-Optics Engineering Department, The Jerusalem College of Technology, Jerusalem 91106, Israel; ilangadasi@gmail.com

**Keywords:** hyper-spectral imaging, push-broom method, birefringence

## Abstract

Hyper-spectral imaging (HSI) systems can be divided into two main types as follows: a group of systems that includes a dedicated dispersion/filtering component whose role is to physically separate the different wavelengths and a group of systems that sample all wavelengths in parallel, so that the separation into wavelengths is performed by signal processing (interferometric method). There is a significant advantage to systems of the second type in terms of the integration time required to obtain a signal with a high signal-to-noise ratio since the signal-to-noise ratio of methods based on scanning interferometry (Windowing method) is better compared to methods based on dispersion. The current research deals with the feasibility study of a new concept for an HSI system that is based on scanning interferometry using the “push-broom” method. In this study, we investigated the viability of incorporating a simple birefringent plate into a scanning optical system. By exploiting the motion of the platform on which the system is mounted, we extracted the spectral information of the scanned region. This approach combines the benefits of scanning interferometry with the simplicity of the setup. According to the theory, a chirped cosine-shaped interferogram is obtained for each wavelength due to the nonlinear behavior of the optical path difference of light in the birefringent plate as a function of the angle. An algorithm converts the signal from a superposition of chirped cosine signals to a scaled interferogram such that Fourier transforming (FT) the interferogram retrieves the spectral information. This innovative idea can turn a simple monochrome camera into a hyperspectral camera by adding a relief lens and a birefringent plate.

## 1. Introduction

The spectral behavior of light reflected from substrates has long been used for characterizing the substrate’s characteristics in agriculture [1], food control [2,3], astronomy [4], medicine [5], security [6], and more. When the spectra of a 2-D field of view are measured simultaneously, a spectral imaging camera or an imaging spectrometer must be used. There are numerous optical designs for realizing a spectral imaging camera or an imaging spectrometer, such as Fourier transform spectrometers (FTS) [7,8,9], dispersive spectrometers [10,11], and others. In recent years, several types of non-scanning FTSs have been introduced [12]. All of these systems are based on complex components, such as the Wollaston prism array [9] and birefringent retarder array [13].

In many cases, there is a relative motion between the object and the measuring optical system. This movement can be exploited to simplify the system and to reduce its cost. A hyperspectral imaging system based on a static polarization interferometer where the relative movement between a scene and the imaging system is exploited is described in [14,15]. In this hyperspectral imaging system, a static polarization interferometer is located on the detector’s array, as shown in Figure 1.

The static polarization interferometer comprises two birefringent wedges, which are located between a polarizer and an analyzer. The orientations of the two optical axes of the birefringent materials, of which the two wedges are made, are perpendicular to each other. In the imaging process, the light from each object point is focused on an image point on the detector plane while propagating through the static polarization interferometer. In the static polarization interferometer, the two orthogonal polarizations of light acquire a phase difference, and after passing through the analyzer, they interfere with the detector plane. The phase difference is a function of the location in which the two orthogonal polarizations propagate through the two wedges of the static polarization interferometer. When there is a relative motion between the object and the measuring optical system, each image point scans the detector’s plane in the direction opposite to the relative motion between the object and the measuring optical system, and the phase difference between the two orthogonal polarization changes. Accordingly, the relative motion encodes the spectral information in the detector’s temporal signal via a “push-broom” approach.

However, in an imaging system, when the light rays of each point of the scene are focused by the imaging lens towards the detectors, they propagate at different angles and, thus, enter the static polarization interferometer at different angles, as described in [16]. Accordingly, there is a change in the phase difference with the azimuthal angle (the plane of incidence relative to the optic axis) and polar angle of incidence. The phase difference changes as a function of the illumination-cone ray angle, determined by the f/number (beams with smaller f/numbers show greater variation across the cone). Extreme rays tend to be less or more retarded than the rays at normal incidence, depending upon the azimuthal angle.

As a result, this effect tends to decrease the spectral resolution of the hyperspectral imaging system, especially in imaging systems with low f/numbers. Moreover, since the spectrometer is required to be before the detector and there is a cone of rays that converge, the diameter at which the rays enter the spectrometer is significantly larger than the sampling pixel size, which reduces the spectral resolution. Another disadvantage is that since it is close to the focal plane, the quality of the components produced and the level requirements are extremely high.

Another study [17] introduced a new concept of a static polarization spectrometer. In this approach, the hyperspectral camera was based on the Savart polariscope. The Savart polariscope consists of two identical birefringent plates. The optical axes of the plates are perpendicular to each other. The Savart polariscope splits the source into two virtual sources in which their light interferes with a detector. The OPD between the two virtual sources at a point on the detector depends on the displacement between the sources. From the relative “push-broom” motion of the device and the source, the position of the point of interference changes on the detector, and an interferogram is obtained. The displacement between the sources depends on the birefringent refractive indices, the thickness of the plates, and the angle of incident to the optical system. The researchers demonstrated the ability to build an interferogram as a function of the incident angle and extraction of the spectrum by FT. This device has the advantages of compact size, a wide field of view, high throughput, and no moving parts. However, since the interferogram is a function of the incident angle, a chirped cosine-shaped interferogram is obtained for each wavelength due to the nonlinear behavior of the optical path difference, and this signal should be scaled. In addition, for a field of view (FOV) less than $6^{{}^{\circ}}$, the OPD increases as a function of the incidence angle, while for an FOV larger than $6^{{}^{\circ}}$, the OPD decreases as a function of the incidence angle. This makes the signal ambiguous for FOV larger than $6^{{}^{\circ}}$.

In this paper, we introduce an instrument based on a simpler setup that uses only one simple birefringent plate, and the need for using the Savart polariscope is eliminated. Similarly to the Savart polariscope system, the path difference in our system is a nonlinear function, which should be scaled by signal processing before performing an FT to extract the spectrum.

## 2. Materials and Methods

### 2.1. Principle of System Operation

Figure 2 describes a schematic 4F hyperspectral imaging system based on the new concept of the static polarization interferometer located in the path of collimated light rays emerging from each object’s point.

This static polarization interferometer comprises a waveplate located between two parallel polarizers, as shown in Figure 3.

The waveplate is a flat plate made of uniaxial birefringent material whose optical axis orientation is parallel to the optical axis of the optical system. The waveplate creates a path of difference between the orthogonal polarizations of light, which varies as a function of the direction of the propagation of the light rays through the waveplate. When a collimated non-polarized light beam propagates through the first polarizer, the light beam is split into two light beams with two orthogonal polarizations. These two beams acquire a phase difference ∆φ whose value is a function of the direction in which the original light non-polarized beam propagates through the waveplate. The waveplate is flat; hence, the two split orthogonal polarization light beams that emerge from the static polarization interferometer are parallel. The second polarizer polarizes the two light beams along its polarization direction.

Since the two orthogonal polarizations light beams are parallel, they are focused by the imaging lens on the same image point at which the detectors array and interfere.

As the image of the scene moves from line to line successively on the detector array of the imaging system in a direction opposite to that of relative motion, the phase shift between the two orthogonal polarizations of light beams varies from line to line at the successive lines of the detectors array. Accordingly, as the position of the point of the interference changes on the detector, a chirped interferogram is obtained. After scaling the interferogram by signal processing and performing FT, the spectrum of each object’s point is extracted.

Using a “push-broom” approach, each point of the scene is tracked to record and extract the signal from which the hyperspectral data cube of the full scene is calculated. However, for all image points their scans do not pass through the optical axis, their rays enter the waveplate at a fixed angle in the axis perpendicular to the scan line, and this results in an additional phase for each ray. Thus, as the object’s point is further away from the optical axis, for a certain wavelength, the successive peaks of the interferogram move to the center of the interferogram. This results in a change in the interferogram received on this detector line compared to the interferogram received on the central detector line and may affect the spectral reconstruction.

#### 2.1.1. Phase Shift Introduced by an Anisotropic Uniaxial Plane-Parallel Plate

The general expression for the optical path difference (OPD) between the ordinary and the extraordinary rays introduced by an anisotropic uniaxial plane-parallel plate with a thickness d is as follows [18]:(1)$$\begin{array}{c} OPD\left(\alpha ,\theta ,\delta ,L,n,n_{o},n_{e}\right)\\ =d(\sqrt{(n_{o}^{2}-n^{2}\sin^{2}\alpha )}+\frac{n\left(n_{o}^{2}-n_{e}^{2}\right)sin\theta cos\theta cos\delta sin\alpha }{n_{e}^{2}\sin^{2}\theta +n_{o}^{2}\cos^{2}\theta }\\ -\frac{n_{o}\sqrt{n_{e}^{2}(n_{e}^{2}\sin^{2}\theta +n_{o}^{2}\cos^{2}\theta )-[n_{e}^{2}-\left(n_{e}^{2}-n_{o}^{2}\right)\cos^{2}\theta \sin^{2}\delta ]n^{2}\sin^{2}\alpha }}{n_{e}^{2}\sin^{2}\theta +n_{o}^{2}\cos^{2}\theta }) \end{array}$$
where $n_{o}$ and $n_{e}$ are the ordinary and the extraordinary refractive indexes of the uniaxial plane-parallel plate immersed in an isotropic medium with refractive index n, θ is the angle between the optical axis and the interface, δ is the angle between the plane of incidence and the optical axis projection on the interface and α is the angle of incidence.

In our system, the waveplate is immersed in the air, and the orientation of the waveplate’s optical axis is parallel to the optical axis of the system; hence, θ=90° and n=1.

In this case, Equation (1) is reduced to the following:(2)$$OPD\left(\alpha ,d,n_{o},n_{e}\right)=d\left(\sqrt{(n_{o}^{2}-\sin^{2}\alpha )}-\frac{n_{o}\sqrt{n_{e}^{2}-\sin^{2}\alpha }}{n_{e}}\right)$$

This equation is used to calculate the OPD between the two orthogonal polarization beams and the signal that is generated by the optical system.

#### 2.1.2. The Interferogram

Assuming that for a certain wavelength, the intensity of the two polarization beams is equal to $I\left(\lambda \right)$, the intensity of the interference between the two polarizations is as follows:(3)$$I(\lambda ,\alpha ,d,n_{o},n_{e})=2\cdot I\left(\lambda \right)\left[1+\cos\left(\frac{2\pi }{\lambda }\cdot OPD(\alpha ,d,n_{o},n_{e})\right)\right]$$

The interferogram is the integral of the interference intensities of all wavelengths, and the spectrogram is obtained by cosine Fourier transforming (CFT) the interferogram [19,20].

### 2.2. Building the Hyperspectral Imaging System in ZEMAX

The hyperspectral imaging system was simulated by ZEMAX OpticStudio 20.1.2 software using the Sequential mode and then converted to the Non-Sequential mode in order to simulate the interference of coherent rays on the detector.

The system consists of a collimation lens, a 2 mm thickness waveplate located between two polarizers, a converging lens, and a detector, as shown in Figure 4. The hyperspectral imaging system images a point light source on the detector plane. To imitate the relative movement between the object and the imaging system, the point light source was translated along a line in the FOV.

The system was designed to work in the 400–700 nm range where the THORLABS AL1225G-A lenses with a focal length f=25 mm, diameter ∅=10.6 mm, and AR350–700 nm coating were used as the collimation and converging lenses.

The waveplate was defined as a Calcite plate with the birefringence indices: $n_{o}=1.663,n_{e}=1.489$ where, in the simulation, the dispersion was ignored.

The light rays from the source were traced, and the complex amplitudes of the traced rays that stroke the same pixel were summed and squared to yield the intensity of the pixel.

## 3. Results

The first step was to compare the interferogram obtained by the ZEMAX simulations and the expected interferogram according to Equations (2) and (3).

Figure 5 shows the comparison between the interferogram obtained by the ZEMAX simulation and the interferogram constructed according to Equations (2) and (3) for the wavelength λ=532nm and a Calcite waveplate with the thickness d=2mm, using the same arbitrary intensity.

It can be seen that a chirped interferogram was obtained, and there is a very good match between the expected interferogram using Equations (2) and (3) and the interferogram obtained by the ZEMAX simulation, which validates the ZEMAX simulation.

Due to the nonlinear dependency of OPD on the incident angle, the frequency of the cosine interferogram produced for each wavelength varies, and a chirped cosine interferogram can be obtained. Figure 6 shows the OPD as a function of the incidence angle on a 10 mm Calcite waveplate.

It is necessary to scale the chirped cosine signals to achieve an accurate spectrogram from the interferogram. This scaling process involves linearizing and interpolating the optical path difference (OPD) to ensure that the chirped interferogram becomes linear and continuous. By applying the CFT to the scaled interferogram, the desired spectrogram can be obtained.

To check the spectrum extraction method, a theoretical interferogram of three wavelengths was constructed using Equation (3). The interferogram consisted of the three wavelengths, 400, 500, and 600 nm, with the same amplitudes, for the following parameters: maximum angle (half field)—10 degrees, waveplate thickness—10 mm, and sampling steps—100 micro-radians. The resulting combined interferogram is shown in Figure 7.

Figure 8 shows the interferogram after scaling, linearizing, and interpolating, where the black dots are the sample points of the original signal.

Figure 9 displays the three wavelengths constructed after applying the FFT algorithm on the scaled interferogram.

As can be seen from the figure, accurate amplitudes and frequencies were reconstructed.

### 3.1. On-Axis and off-Axis Spectrum Extraction

The on-axis and off-axis spectrum extraction were investigated. Two simulations were performed using ZEMAX as follows: one to extract the spectrogram on-axis and the other to extract the spectrogram off-axis.

#### 3.1.1. On-Axis Spectrum Extraction

In the on-axis simulation, the object was placed on the optical axis of the system, and in the off-axis simulation, the object was placed off-axis from the optical axis of the system.

Using the ZEMAX, the interferogram of two wavelengths $\lambda_{1}=532\;nm$ and $\lambda_{2}=700\;nm$ with a Calcite waveplate and a thickness of 10 mm was created. The point light source was placed in the center of the FOV and was translated along a line from the optical axis up to the height of 2.2 mm from the optical axis, which is equivalent to the maximum FOV angle (half field) of $5.02^{{}^{\circ}}$. The signal obtained for the two wavelengths and the interferogram (their sum) as a function of the angle of incidence are shown in Figure 10. As shown, there was a decrease in the intensities of the interferograms at higher angles due to the “cosine fourth law” reflection on the waveplate aberrations.

After scaling and applying the CFT on the scaled interferogram, two frequencies were obtained as expected. Figure 11 shows a comparison between the expected and the resulting frequencies. It can be seen that the two resulting frequencies are similar to the theoretical frequencies. However, the difference between the resulting intensities and the expected intensities was 14%. After correcting the decrease in the intensities of the interferograms at higher angles, the difference between the resulting intensities and the expected intensities was reduced to 3%.

#### 3.1.2. Off-Axis Spectrum Extraction

The interferogram obtained for an off-axis object’s point differs from that obtained for an on-axis object’s point due to the initial OPD between the two polarizations, which depends on the incident angle.

To demonstrate the effect of the off-axis object placement on the interferogram, two simulations were performed using ZEMAX. In both simulations, the same parameters were used as in the on-axis case. However, in the off-axis simulations, the point light source was moved off the center of the FOV along two lines. This caused the rays from the object’s point to intercept the waveplate at two different off-axis angles of 2 degrees and 10 degrees.

Figure 12 shows a comparison of the reconstructed frequencies obtained using the ZEMAX interferograms to the frequencies of the corresponding theoretical signal. The results show that the reconstructed frequencies of the different wavelengths are identical, regardless of the location of the object point relative to the optical axis. However, as the row moves farther from the center, the relationship between the intensities of the different wavelengths changes due to the initial OPD between the two polarizations and due to the decrease in the intensities of the interferograms at higher angles.

The simulations showed that within an FOV of up to $10^{{}^{\circ}}$, the errors due solely to the initial OPD between the two polarizations are limited to below ±3% for all frequencies in the range of 500–700 nm. If the interferogram measurement is extended to $20^{o}$ while the off-axis spectral measurements are limited to $10^{o}$, the errors reduce to below ±1%. The interferogram obtained with the initial OPD between the two polarizations at an increasing off-axis angle can be understood as an interferogram that lacks the information between the zero OPD and the initial OPD. Accordingly, the off-axis spectral measurement angle may be increased by interpolating the interferogram as the frequency of its ripple is known.

### 3.2. Spectral Resolution

It is well known that the spectral resolution Δλ of FTS is given as follows:(4)$$\Delta \lambda \approx \frac{{\bar{\lambda }}^{2}}{{OPD}_{max}}$$
where $\bar{\lambda }$ is the mean wavelength.

Accordingly, for the given refractive indices $n_{o}$ and $n_{e}$, the maximum angle of incidence on the waveplate α and spectral resolution Δλ, the required waveplate thickness d is as follows:(5)$$d_{waveplate}\left(\Delta \lambda ,{\bar{\lambda }}^{2},n_{o},n_{e},\alpha \right)=\frac{{\bar{\lambda }}^{2}}{\Delta \lambda \cdot \left(\sqrt{(n_{o}^{2}-\sin^{2}\alpha )}-\frac{n_{o}\sqrt{n_{e}^{2}-\sin^{2}\alpha }}{n_{e}}\right)}$$

Figure 13 shows the required waveplate thickness as a function of the spectral resolution for optical systems with an FOV of 5, 10, 15, and 20 degrees for $\bar{\lambda }=0.5\mu m$.

In the visible spectrum, it is feasible to incorporate this concept into a practical imaging system by utilizing a waveplate with a thickness ranging from several millimeters to centimeters. The optimal thickness depends on the system’s FOV and the desired spectral resolution. For instance, a waveplate with a thickness of approximately 12 mm enables a spectral resolution of 10 nm in a system with a horizontal FOV of 10 degrees. This demonstrates the potential for implementing this concept in real-world imaging applications, providing enhanced spectral information for various imaging tasks.

### 3.3. Sampling Resolution

The number of pixels in the detector determines the sampling resolution of the signal. Figure 14 illustrates the FT of the signal for a system with the following parameters: FOV of 10 degrees, waveplate thickness of 11.75 mm, and three wavelengths of 490, 500, and 510 nm. The signal was sampled with varying sampling steps as follows: 1 mrad, 500 μrad, and 100 μrad, corresponding to 350, 700, and 3500 pixels, respectively.

The results indicate that, for this system, there is no discernible difference in the spectrum obtained for the sampling steps of 100 μrad and 500 μrad. However, when the sampling step is increased to 1 mrad, undersampling occurs, resulting in a degraded reconstructed spectrum. This demonstrates the importance of selecting an appropriate sampling step to ensure accurate spectral reconstruction.

## 4. Conclusions

We propose a hyperspectral imaging system based on the “push-broom” method, which incorporates a simple birefringent plate into a scanning optical system. By exploiting the movement of the platform on which the system is installed, spectral information can be extracted from the scanned area. The feasibility of this approach was demonstrated through an analysis performed using ZEMAX software.

The integration of a birefringent plate into the optical setup allows for the generation of chirped cosine-shaped interferograms for each wavelength. To retrieve the spectral information, we developed an algorithm capable of converting the composite chirped cosine signals into a scaled interferogram. The application of FT into this interferogram enables the extraction of the desired spectral information.

Furthermore, our investigation involved examining the achievable resolution using this method and determining the necessary plate thickness to achieve specific resolutions. These considerations play a crucial role in optimizing the performance of the hyperspectral imaging system, ensuring that it meets the requirements of various applications.

## Figures and Tables

**Figure 1 sensors-24-03092-f001:**
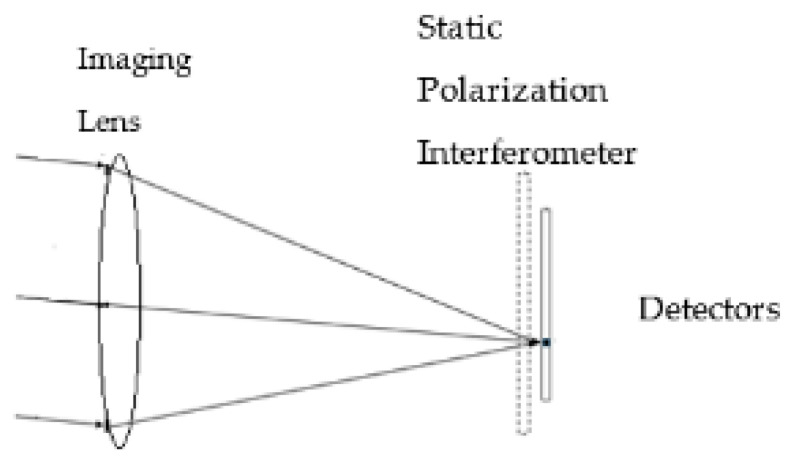
Hyperspectral imaging system based on a static polarization interferometer.

**Figure 2 sensors-24-03092-f002:**
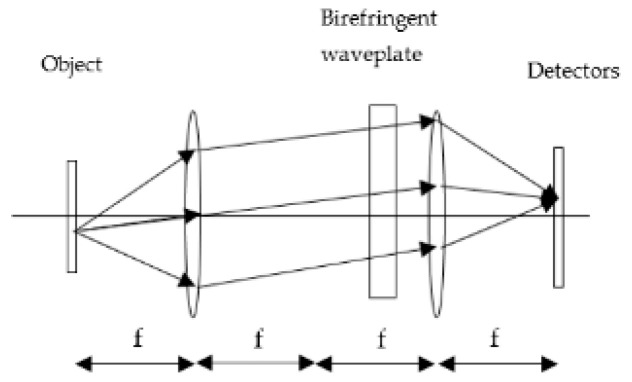
A 4F hyperspectral imaging system based on a static polarization interferometer in which the refractive index difference of the two polarizations changes as a function of the angle of the rays.

**Figure 3 sensors-24-03092-f003:**
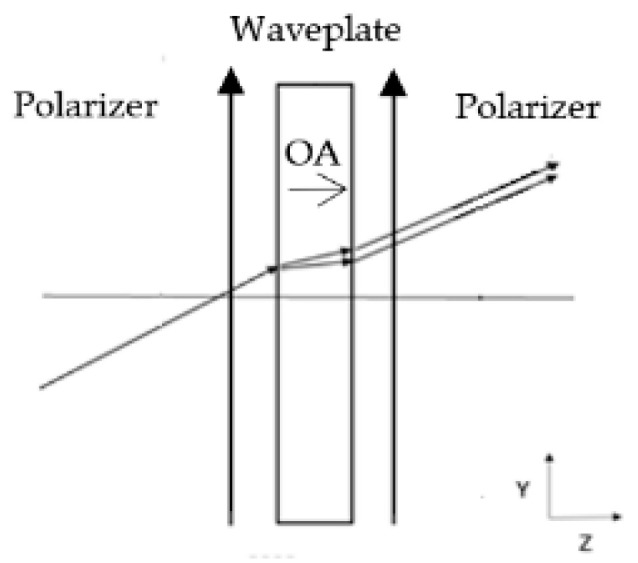
A waveplate located between two parallel polarizers. The waveplate is made of birefringent material whose optical axis orientation is parallel to the optical axis of the optical system.

**Figure 4 sensors-24-03092-f004:**
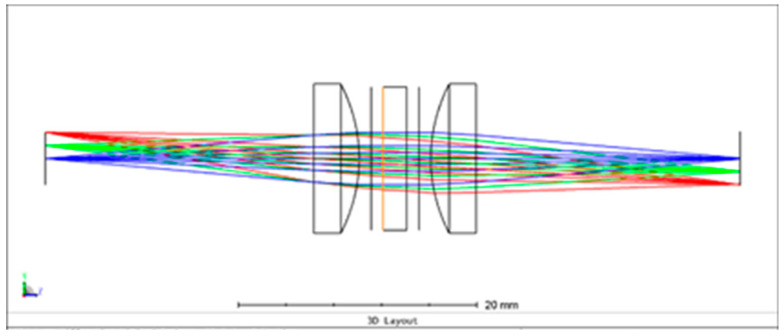
The optical system in ZEMAX. The different colors represent rays from different locations of the point light source.

**Figure 5 sensors-24-03092-f005:**
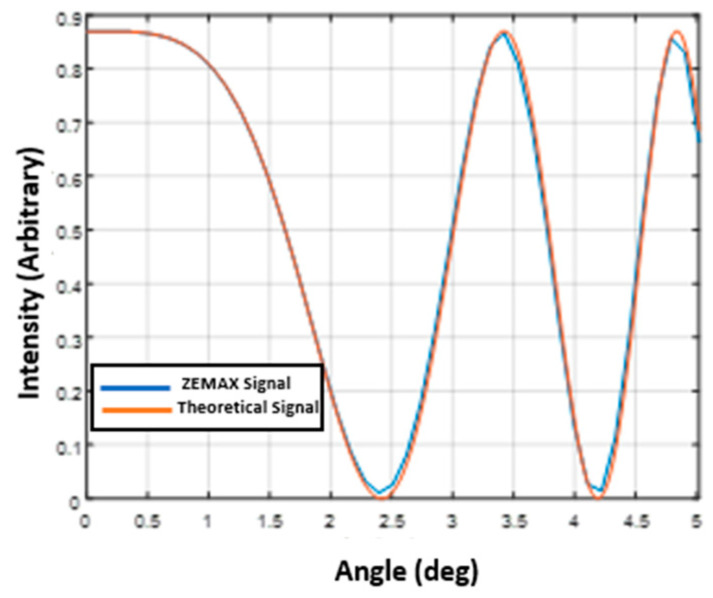
Comparison between the chirped expected interferogram and the interferogram obtained by the ZEMAX simulation for the wavelength λ=532 nm and a Calcite waveplate with the thickness =2 mm.

**Figure 6 sensors-24-03092-f006:**
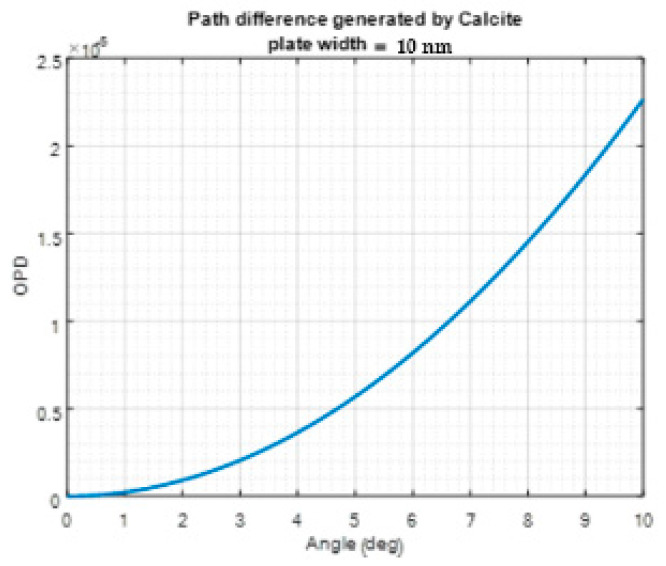
The OPD obtained as a result of passing at different angles through a 10 mm thick Calcite waveplate.

**Figure 7 sensors-24-03092-f007:**
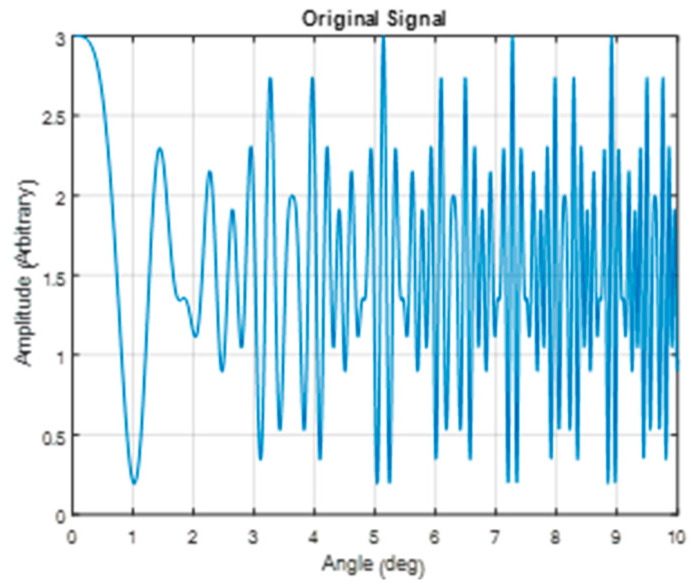
Theoretical combined interferogram of three wavelengths—400, 500, 600 nm—as a function of the incidence angles.

**Figure 8 sensors-24-03092-f008:**
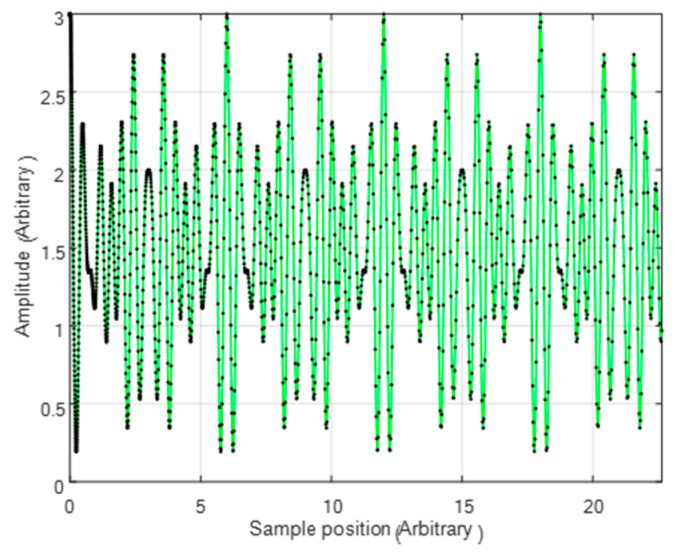
The scaled interferogram where the black dots are the sample points of the original signal and the green line is the interpolated interferogram.

**Figure 9 sensors-24-03092-f009:**
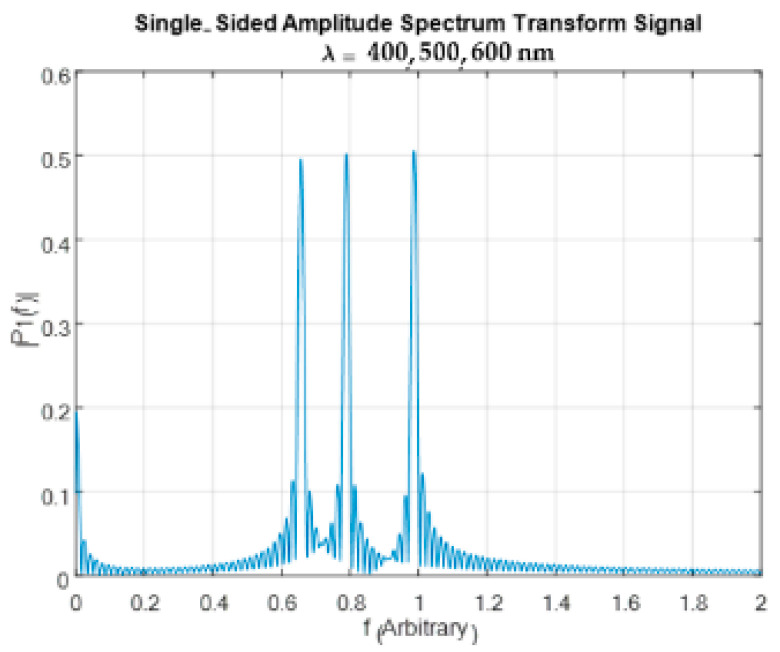
The three wavelengths constructed after applying the FFT algorithm on the scaled signal.

**Figure 10 sensors-24-03092-f010:**
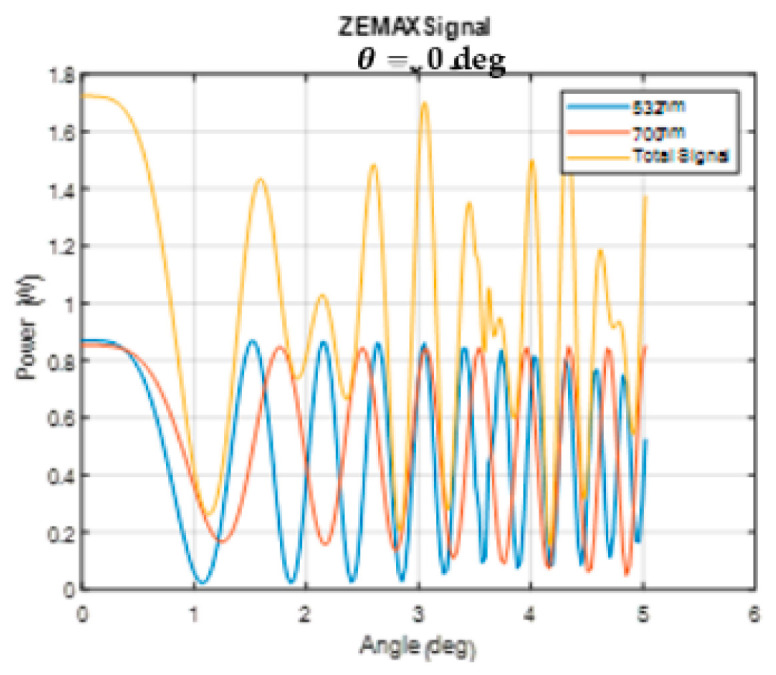
The ZEMAX signals for two wavelengths $\lambda_{1}=532nm$ and $\lambda_{2}=700nm$ for a 10 mm Calcite waveplate thickness.

**Figure 11 sensors-24-03092-f011:**
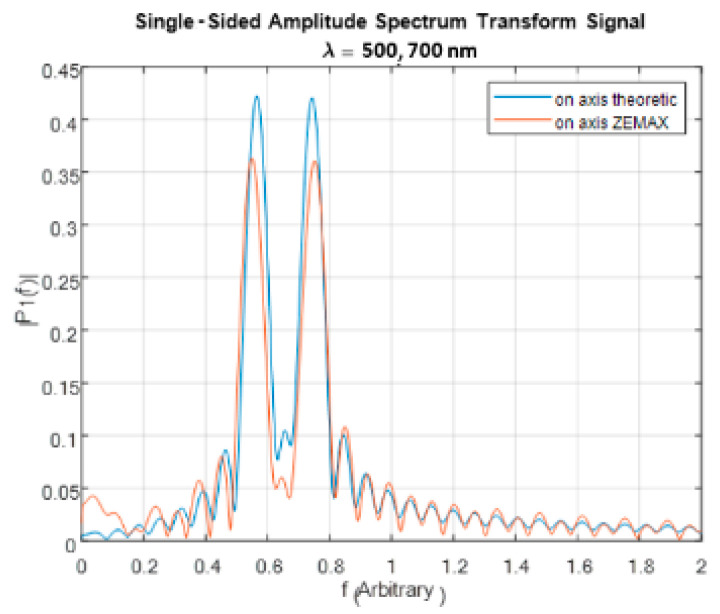
The frequencies reconstructed from the ZEMAX interferogram and the corresponding theoretical interferogram.

**Figure 12 sensors-24-03092-f012:**
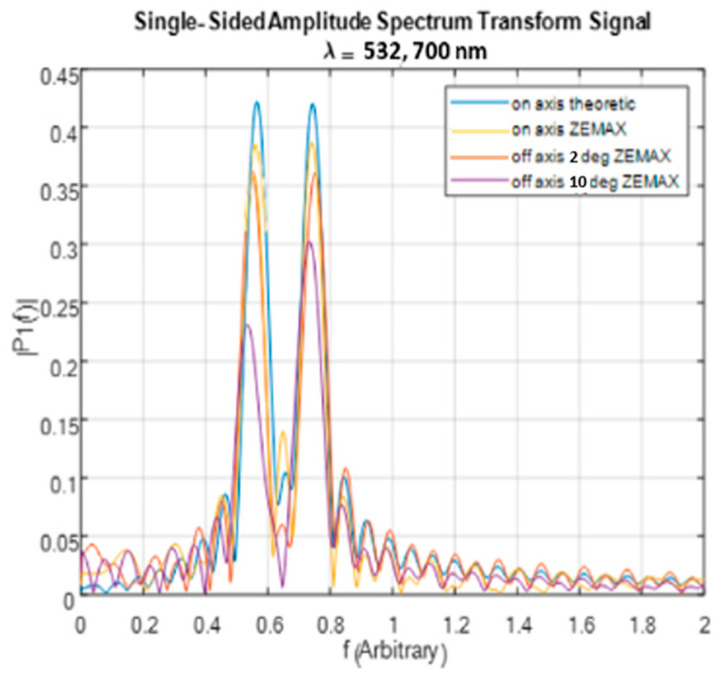
Comparison of reconstructed frequencies from the ZEMAX interferogram and theoretical frequencies for the off-axis case.

**Figure 13 sensors-24-03092-f013:**
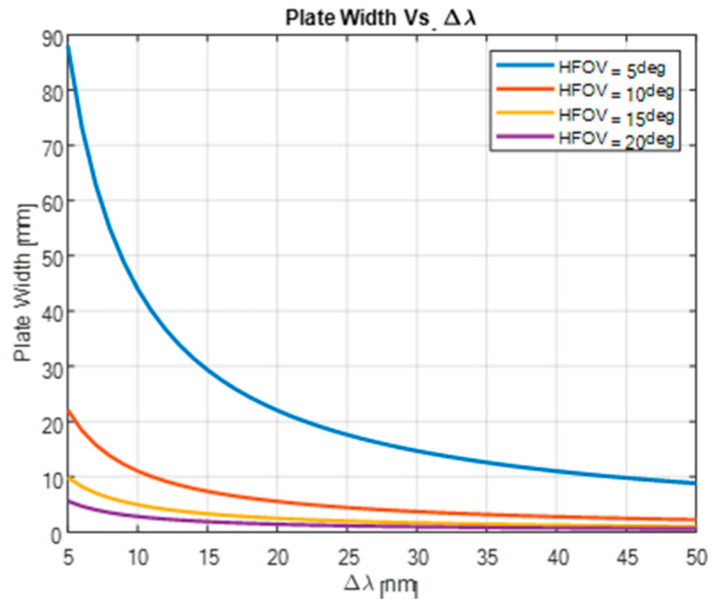
Waveplate thickness as a function of the spectral resolution for 5, 10, 15, and 20 degrees FOV.

**Figure 14 sensors-24-03092-f014:**
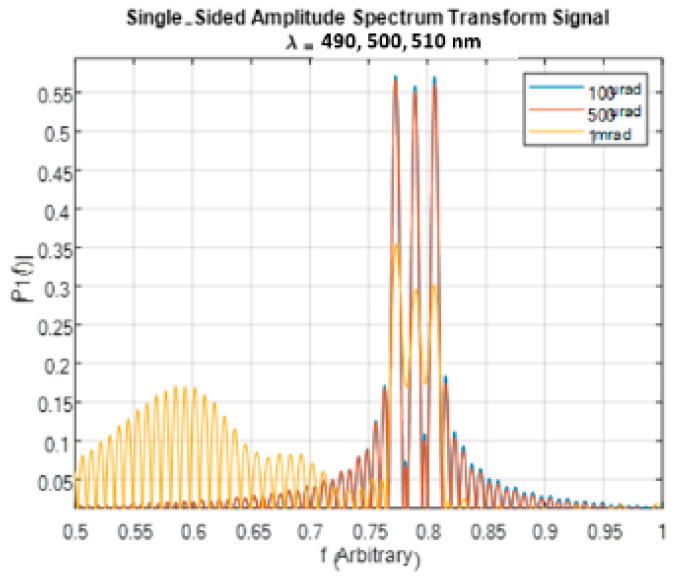
FT of a theoretical signal for different sampling steps.

## Data Availability

No new data were created or analyzed in this study. Data sharing is not applicable to this article.

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
