# Peer review of "Feasibility Study of Scanning Spectral Imaging Based on a Birefringence Flat Plate"

_sensors, 2024, doi:10.3390/s24103092_

Round 1
Reviewer 1 Report
Comments and Suggestions for Authors
I propose that the title will include the words "Feasibility study", since this is a feasibility study and the concept still was not proved in the lab. yet.
Additional comments
- This paper shows the feasibility to design and to make a simple push-broom hyperspectral system by taking a simple monochrome camera and adding a relief lens and a birefringent plate in its optical path?
- part 2.1 that shows the optical layout of the system with the birefringent plate the heart of the system. This design is a first step toward simple push-broom miniature hyperspectral system.
- The authors add a plate of birefringent crystal and embedded if in the optical path in a way that minimized the overall systems.
- The main improvement is a combination of single bi-refringent in the optical path along with the correct algorithms to minimize the chirps.
- The work is a theoretical/simulation work, the conclusions reflect in general idea and the concept of such system that are based on chirp free system, that its output is an interferogram that can be transalated using a simple furrier transform to obtain the full spectral information.
- All references are appropriate
- The Several typos in the text, English editing is recommended.
My recommendations is to accept with minor changed.
End Comments
Author Response
Thank you very much for taking the time to review this manuscript. Please find the detailed responses to you comments below and a clean version of the corrected manuscript. In the revision, I have included the line numbers of the old version and the line numbers in the new version where I have made changes or additions.
Review
- Comment:
I propose that the title will include the words "Feasibility study", since this is a feasibility study and the concept still was not proved in the lab. yet.
Answer:
We have added the words "Feasibility study" in the title.
- Comment:
The Several typos in the text, English editing is recommended.
Answer:
We have corrected the typos:
- Old line 35: "0" to "Figure 1". New line 44.
- Old line 100: "Fig." to "Figure". New line 107.
- Old line 116: "0" to "Figure 3". New line 124.
- Old line 119: "Poalarizer", "Analaizer" to "Polarizer", "Polarizer". New line 127.
- Old line 130: " a polarizer and an analyzer" to " two parallel polarizers". New line 136.
- Old line 141: "analyzer " to " The second polarizer ". New line 147.
- Old line 190: "Fig." to "Figure". New line 198.
- Old lines 198, 208, 212, 220, 222, 255, 261: "calcite" to "Calcite". New lines 207, 217, 225, 233, 234, 274, 283.
- Old line 216 " A. Scaling" was deleted. New line 230.
- Old line 219: "Error! Reference source not found." To "Figure 6". New line 232.
- Old line 233 "0" to "Figure 7". New line 245.
- Old line 237 "0" to "Figure 8". New line 256.
- Old line 237 "interpulating" to "interpolating". New line 256.
- Old line 241 "Fig." to "Figure". New line 260.
- Old line 259 "Fig." to "Figure". New line 278.
- Old line 264: "0" to "Figure 11". New line 286.
- Old line 268: " frequncies" to " frequencies ". New line 306.
- Old line 279 "Fig." to "Figure". New line 317.
- Old line 298 "Fig." to "Figure". New line 349.
- Old line 301 " resoltion " to " resolution". New line 352.
- Old line 317 " 13 " to " 14". New line 368.

Reviewer 2 Report
Comments and Suggestions for Authors
In this manuscript, the author proposed a Hyper-spectral imaging system based on the push-broom method. A birefringent plate was used to generate the optical path difference between the two orthogonal polarizations beams. The spectral information was obtained by analyzing the scaled interferogram. The author proved the feasibility of the imaging system through simulation results.
Here are some questions to the author:
1. The author mentioned the challenges of Hyper-spectral imaging systems in the abstract, such as SNR, sensitivity to component movements. However, the author did not analyze the SNR, sensitivity of the new design. Please rewrite the abstract and highlight the innovation of the paper.
2. The author uses different expressions to describe the same phrase in the article, such as Fourier Transform Spectrometers in upper and lower case. Please define the abbreviation of the phrase the first time it occurs and use the abbreviated form in the rest of the paper.
3. The text in Figure 1, 2, 6, and 9 is partially blocked, please redraw these figures. Meanwhile, Figure x is written as 0 several times in the paper (line 35, 116, 232, 236). Please correct it.
4. Is the waveplate between two polarizers or between polarizer and analyzer? Please indicate this in Figure 3.
5. Please provide the model of Thorlabs lens used in the simulation, or give Ø, f and whether it is coated. For the waveplate, what material is used and how are n0 and ne obtained?
6. What is “A. A. A. Scaling” in line 215?
7. The error appearing on line 218 should be Figure 6.
8. Duplicate content appears in lines 319-323, please correct it.
Comments on the Quality of English LanguageThere are grammatical errors in the paper, please correct it.
Author Response
Thank you very much for taking the time to review this manuscript. Please find the detailed responses to your comments below and a clean version of the corrected manuscript. In the revision, I have included the line numbers of the old version and the line numbers in the new version where I have made changes or additions.
Review
- Comment:
The author mentioned the challenges of Hyper-spectral imaging systems in the abstract, such as SNR, sensitivity to component movements. However, the author did not analyze the SNR, sensitivity of the new design. Please rewrite the abstract and highlight the innovation of the paper.
Answer:
We have rewritten the abstract: Old lines 7-11 to new lines 8-20.
- Comment:
The author uses different expressions to describe the same phrase in the article, such as Fourier Transform Spectrometers in upper and lower case. Please define the abbreviation of the phrase the first time it occurs and use the abbreviated form in the rest of the paper.
We have corrected:
Old lines: 14, 26, 27, 86, 96, 150, 184,227, 263, 290, 312, 317, 334
New lines: 23, 35, 36, 92, 103, 156, 191, 239, 285, 339, 363, 368, 383
- Comment:
The text in Figure 1, 2, 6, and 9 is partially blocked, please redraw these figures. Meanwhile, Figure x is written as 0 several times in the paper (line 35, 116, 232, 236). Please correct it.
Answer:
- The text in Figure 1, 2, 6, and 9 was corrected.
- Old line 35: "0" to "Figure 1". New line 44.
- Old line 116: "0" to "Figure 3". New line 124.
- Old line 233 "0" to "Figure 7". New line 245.
- Old line 237 "0" to "Figure 8". New line 256.
- Old line 264: "0" to "Figure 11". New line 286.
- Comment:
Is the waveplate between two polarizers or between polarizer and analyzer? Please indicate this in Figure 3.
Answer:
We have corrected:
- Old line 119: "Poalarizer", "Analaizer" to "Polarizer", "Polarizer". New line 127.
- Old line 130: " a polarizer and an analyzer" to " two parallel polarizers". New line 136.
- Old line 141: "analyzer " to " The second polarizer ". New line 147.
- Comment:
Please provide the model of Thorlabs lens used in the simulation, or give Ø, f and whether it is coated. For the waveplate, what material is used and how are n0 and ne obtained?
Answer:
- We added the details of the lenses: line 204-205.
- The material is Calcite, lines 207, 217, 225, 233, 234, 274, 283.
- Comment:
What is “A. A. A. Scaling” in line 215?
Answer:
Old line 216 "A. A. Scaling" was deleted. New line 230.
- Comment:
The error appearing on line 218 should be Figure 6.
Answer:
Old line 219: "Error! Reference source not found." Corrected to "Figure 6". New line 232.
- Comment:
Duplicate content appears in lines 319-323, please correct it.
Answer:
Deleted.
- Comment:
There are grammatical errors in the paper, please correct it.
We have corrected the typos:
- Old line 100: "Fig." to "Figure". New line 107.
- Old line 119: "Poalarizer", "Analaizer" to "Polarizer", "Polarizer". New line 127.
- Old line 130: " a polarizer and an analyzer" to " two parallel polarizers". New line 136.
- Old line 141: "analyzer " to " The second polarizer ". New line 147.
- Old line 190: "Fig." to "Figure". New line 198.
- Old line 237 "interpulating" to "interpolating". New line 256.
- Old line 241 "Fig." to "Figure". New line 260.
- Old line 259 "Fig." to "Figure". New line 278.
- Old line 268: " frequncies" to " frequencies ". New line 306.
- Old line 279 "Fig." to "Figure". New line 317.
- Old line 298 "Fig." to "Figure". New line 349.
- Old line 301 " resoltion " to " resolution". New line 352.
- Old line 317 " 13 " to " 14". New line 368.

Reviewer 3 Report
Comments and Suggestions for Authors
The manuscript proposes a hyperspectral imaging system based on the "push-broom" method, which incorporates a simple birefringent plate into a scanning optical system. By exploiting the movement of the platform on which the system is installed, spectral information can be extracted from the scanned area. Overall, the manuscript provides an idea can turn a simple monochrome camera into a hyperspectral camera by adding a relief lens and a birefringent plate. However, the following issues need to be addressed before considering whether or not to admit a manuscript:
1、There are many formatting errors in the manuscript: the text of the pictures is incomplete, for example, Fig. 1, Fig. 2, Fig. 10, Fig. 11, Fig13 etc.; the position of the pictures is not centred and many of them are wrongly cited in Lines 35, 115, 232 etc.; the formulae are incorrectly numbered, Lines 181, etc. Besides, the titles of the literature cited in the article should be written in the form [1], [2] instead of 1, 2.
2、The meaning of line 215 in the manuscript is unknown; the meaning of lines 218-219 is unknown.
3、 Figure 7 shows the theoretical interference intensity at three different wavelengths, 400nm, 500nm and 600nm, without differentiation.
4、The manuscript proposes a way to reflect the spectral information of the light source using the flat plate interference phenomenon, but the accuracy of the results is not reflected in the process of analysing the results, and the process of analysing the interferograms after the Fourier transform should be included.
5、The manuscript mentions off-axis spectral extraction simulation tests, but lacks a quantitative description of their effect on the extraction of spectral information.
Comments on the Quality of English LanguageThe overall English language quality of the manuscript is good, with a small amount of English language requiring editing.
Author Response
Thank you very much for taking the time to review this manuscript. Please find the detailed responses to your comments below and a clean version of the corrected manuscript. In the revision, I have included the line numbers of the old version and the line numbers in the new version where I have made changes or additions.
Review
- Comment:
There are many formatting errors in the manuscript: the text of the pictures is incomplete, for example, Fig. 1, Fig. 2, Fig. 10, Fig. 11, Fig13 etc.; the position of the pictures is not centred and many of them are wrongly cited in Lines 35, 115, 232 etc.; the formulae are incorrectly numbered, Lines 181, etc. Besides, the titles of the literature cited in the article should be written in the form [1], [2] instead of 1, 2.
Answer:
- We have corrected the text of the pictures.
- The position of the pictures is not centred.
- We have corrected the citing of the images:
- Old line 35: "0" to "Figure 1". New line 44.
- Old line 116: "0" to "Figure 3". New line 124.
- Old line 233 "0" to "Figure 7". New line 245.
- Old line 237 "0" to "Figure 8". New line 256.
- Old line 264: "0" to "Figure 11". New line 286.
- We have corrected the titles of the literature.
- Comment:
The meaning of line 215 in the manuscript is unknown; the meaning of lines 218-219 is unknown.
Answer:
- We added a sentence in line 228 (old 215).
- Old line 219: "Error! Reference source not found." Was corrected to "Figure 6" in new line 232.
- Comment:
Figure 7 shows the theoretical interference intensity at three different wavelengths, 400nm, 500nm and 600nm, without differentiation.
Answer:
It is the combined interferogram as explained in line 245.
- Comment:
The manuscript proposes a way to reflect the spectral information of the light source using the flat plate interference phenomenon, but the accuracy of the results is not reflected in the process of analyzing the results, and the process of analyzing the interferograms after the Fourier transform should be included.
Answer:
The accuracy of the results was included:
- In lines 278-280 we explained the reasons for the errors.
- In lines 288-291 we added the analyzed interferogram accuracy after the Fourier transform.
- Comment:
The manuscript mentions off-axis spectral extraction simulation tests, but lacks a quantitative description of their effect on the extraction of spectral information.
Answer:
We added lines 320-332 where we explained the reasons for the inaccuracy in off-axis spectral extraction. We have analyzed the interferogram accuracy after the Fourier transform by simulations and added the results.

Round 2
Reviewer 2 Report
Comments and Suggestions for Authors
All problems have been solved. This manuscript can be considered for publication.
Reviewer 3 Report
Comments and Suggestions for Authors
None